# Novel Peptide-Based Inhibitors for Microtubule Polymerization in *Phytophthora capsici*

**DOI:** 10.3390/ijms20112641

**Published:** 2019-05-29

**Authors:** Sang-Choon Lee, Sang-Heon Kim, Rachel A. Hoffmeister, Moon-Young Yoon, Sung-Kun Kim

**Affiliations:** 1Department of Chemistry, Georgia State University, Atlanta, GA 30303, USA; 2Department of Chemistry and Research Institute for Natural Sciences, Hanyang University, Seoul 04763, Korea; konasi2@naver.com; 3Department of Natural Sciences, Northeastern State University, Tahlequah, OK 74464, USA; hoffmeir@nsuok.edu

**Keywords:** phytophthora blight, phytophthora capsici, phage display, peptide, microtubules

## Abstract

The plant disease Phytophthora blight, caused by the oomycete pathogen *Phytophthora capsici*, is responsible for major economic losses in pepper production. Microtubules have been an attractive target for many antifungal agents as they are involved in key cellular events such as cell proliferation, signaling, and migration in eukaryotic cells. In order to design a novel biocompatible inhibitor, we screened and identified inhibitory peptides against alpha- and beta-tubulin of *P. capsici* using a phage display method. The identified peptides displayed a higher binding affinity (nanomolar range) and improved specificity toward *P. capsici* alpha- and beta-tubulin in comparison to *Homo sapiens* tubulin as evaluated by fluorometric analysis. One peptide demonstrated the high inhibitory effect on microtubule formation with a nanomolar range of IC_50_ values, which were much lower than a well-known chemical inhibitor—benomyl (IC_50_ = 500 µM). Based on these results, this peptide can be employed to further develop promising candidates for novel antifungal agents against Phytophthora blight.

## 1. Introduction

The Phytophthora blight caused by *Phytophthora capsici* (*P. capsici*) is a devastating plant disease primarily affecting the agricultural production of peppers and other economically important vegetable crops worldwide [1,2]. *P. capsici* is a filamentous oomycete pathogen with a broad host range and characteristics that affect several plant species, (e.g., tomato and pepper). *P. capsici* infection causes root rot, leaf and stem blight, and other conditions [1]. Several strategies, such as the rotation of crops and the use of agricultural chemicals, have been widely adopted to control this disease [3,4,5,6]. The fungicides metalaxyl, fosetyl-aluminum, tetrazole, and propamocarb-hydrochloride were found to be quite effective against *P. capsici* infection [7]. These chemicals, however, adversely impact human health, especially children’s health, and adversely impact the environment. Extensive use of these fungicides has led to the emergence of resistance in *P. capsici* [8,9]. Thus, a biologically safe alternative is necessary for the control of Phytophthora blight.

Microtubules are a dynamic polymer composed of α- and β-tubulin (α, β-tubulin) proteins. The α, β-tubulin heterodimers polymerize in a GTP-dependent manner to form microtubules. Historically, microtubules have been recognized as an attractive target in the design of many anticancer and antifungal agents, as these proteins are involved in key cellular events including cell division, cell proliferation, trafficking, signaling, and migration in eukaryotic cells [10].

In the past decade, several new chemical classes which display potent antifungal activity targeting fungal α, β-tubulin have been developed such as zoxamide, inhibiting β-tubulin in *Phytophthora sojae* (*P. sojae*); methyl benzimidazole carbamate, (MBC) inhibiting β-tubulin in *Podosphaera xanthii* (*P. xanthii*); and benomyl [11,12,13,14]. One of these microtubule-targeted fungicides is benomyl. In the presence of benomyl, the polymerization of tubulin to microtubules is inhibited resulting in retarded cell division [15]. Like most fungicides, due to its low solubility, only a small amount of benomyl will dissolve in water [16]. This low solubility inhibits absorption by crops, thus limiting the agricultural application of benomyl [17]. This necessitates the development of highly soluble microtubule-based antifungal agents with high combining ability to α, β-tubulin. The use of such peptides, as inhibitors, may be an attractive alternative in overcoming the limitations associated with small molecule-based antifungal agents. These peptide-based inhibitors enable binding of target molecules with high affinity, specificity, and stability. These inhibitors also allow for easy chemical modification and have low immunogenicity. In addition, such peptide-based inhibitors may express lower toxicity than other chemical-based drugs [18]. In the screening of potential peptides, phage display methods are widely used. These phage display methods include a biopanning technique that can be used to study intermolecular interactions [19]. Using M13 bacteriophage-encoded five copies of the 12-mer peptide, on the pIII minor protein, specific target peptides are screened. These diagnostic and therapeutic probes would be a useful tool in the identification of numerous diseases [20,21,22,23].

In this study, screening and identification of novel peptide-based inhibitors were performed using α, β-tubulins of *P. capsici* as targets. The candidate peptides demonstrated substantial inhibitory potency with high binding affinity and specificity toward *P. capsici* α, β-tubulins. Based on these results, biocompatible peptide-based inhibitors for *P. capsici* α, β-tubulins may be promising candidates in the development of potent agents against Phytophthora blight. 

## 2. Results

### 2.1. Purification of P. Capsici and H. Sapiens α, β-Tubulins

The recombinant α, β-tubulin proteins from *P. capsici* and *H. sapiens* were purified individually to homogeneity using Ni-NTA sepharose affinity chromatography. Analysis of the purified recombinant proteins was confirmed on sodium dodecyl sulfate–polyacrylamide gel electrophoresis (SDS-PAGE) under reducing condition, and the α, β-tubulins for *P. capsici* showed single bands at approximately 55 kDa for α-tubulin and 57 kDa for β-tubulin. Similar sizes of protein bands were observed for the homologous α, β-tubulins from *H. sapiens* (Figure 1). The purified proteins were used to check tubulin activities and screen peptide inhibitors.

### 2.2. The P. Capsici α, β-Tubulin Activities via Polymerization

To confirm the full activities of the purified recombinant *P. capsici* α, β-tubulins, a polymerization assay was performed using the method described in the Materials and Methods section. Polymerization of α, β-tubulins was rapidly initiated upon the addition of GTP in a time-dependent manner, as shown in Figure 2. The rate and extent of the polymerization reaction were monitored by measuring the turbidity at 350 nm (Figure 2A), and the polymerization reaction reached saturation after 40 min of incubation. The possibility of a decrease in band intensity of each α- and β-tubulin was explored using a 12% SDS polyacrylamide gel (Figure 2B). The band intensities of α, β-tubulin progressively decreased with an increase in reaction time due to combining of both proteins, indicating that they were consumed in microtubule formation. It should be noted here that, due to the larger size of the polymerized microtubule assembly, the corresponding bands of microtubules could not be visualized under the assay conditions. 

### 2.3. Screening of P. Capsici α, β-Tubulin Binding Peptide via Phage Display and Determination of Its Binding Affinity and Inhibition

Phage display was conducted with the M13 phage library in an effort to identify specific binding peptides to α, β-tubulin proteins from *P. capsici*. After five rounds of biopanning toward each target protein, 169 phage clones for α-tubulin and 112 for β-tubulin were selected. Subsequently, the DNA sequences of the displayed peptides were determined, and based on the DNA sequences, 12-mer amino acids were translated. As shown in Table 1, a total of 15 different peptides were determined, where 13 different sequences for α-tubulin and two different sequences for β-tubulin were identified. Testing was done for all the phages with different sequences to estimate their binding affinities for α, β-tubulins of *P. capsici*. The binding affinities of phages were found to be in the low picomolar (10^−12^) range, as determined by the enzyme-linked immunosorbent assay (ELISA) method with anti-M13 phage antibody. The results are summarized in Table 1.

As a next step, all 15 phages were used to determine whether the phages could inhibit the activity of microtubule polymerization of *P. capsici*. The concentration of each phage was constant at 10 nM, and the polymerization was measured after 40 min of reaction time. Inhibition was detected by eight phages, while seven phages exhibited a lack of inhibition. The resulting percent inhibitions are shown in Figure 3, where the phage number 1 against α-tubulin (abbreviated as α_P1) showed 28% inhibition, α_P2 showed 54%, α_P3 32%, α_P4 19%, α_P5 5%, α_P5 4%, β_P1 21%, and β_P2 2%.

### 2.4. Binding Affinity and Specificity Evaluation of Synthetic Peptides

To evaluate the binding affinity of selected peptides to target proteins, α_P1, α_P2, α_P3, and β_P1 were chosen for peptide synthesis because these peptides showed promising inhibitory properties. These peptides were synthesized with FITC-label for detection of the difference in the fluorescence by the absence and presence of target binding using a fluorescence microplate reader (see the Materials and Methods). The binding affinities of the selected peptides toward α, β-tubulin proteins were found to be in the low nanomolar range: 69.5 ± 6.4 nM, 93.9 ± 11.2 nM, 79.2 ± 6.8 nM, and 49.7 ± 2.9 nM for α_P1, α_P2, α_P3, and β_P1, respectively (Table 2).

Investigation of the binding affinity for α, β-tubulins of *H. sapiens* to the synthesized peptides is necessary because peptide binding to *H. sapiens* tubulins could trigger severe toxicity in humans. First, α, β-tubulins of *H. sapiens* were individually overexpressed and purified (Figure 1). Similarly, the binding affinities were examined, and the binding of the four peptides was found to be significantly lower for α, β-tubulins of *H. sapiens* than the case for α, β-tubulins of *P. capsici* (Figure 4). In addition, bovine serum albumin (BSA) was used as another control to confirm the specificity of the synthesized peptides. No significant binding fluorescence intensities for BSA were detected. These observations suggest that the identified peptides exhibited species-specific binding and would be safe for use as antifungal agents against Phytophthora blight.

### 2.5. Peptide Inhibitory Effects in Microtubule Assembly

The IC_50_ values of the four synthetic peptides were determined by fitting the relative inhibition data via the below Equation (1), where V_0_ is the reaction rate without inhibitor, V_f_ is the rate at saturation inhibition, and [I] is the inhibitor concentration.
(1)$$V=\frac{\left(V_{0}-V_{f}\right)\times {\mathrm{IC}}_{50}}{{\mathrm{IC}}_{50}+\left[I\right]}+V_{f}\;$$

The inhibition data for the four peptides are in reasonably good agreement with the theoretical curve by the Equation (1), showing that the microtubule formations were inhibited in a dose-dependent manner. From the data, the IC_50_ values were determined to be 2.69 μM, 802 nM, 1.24 μM, and 6.91 μM for α_P1, α_P2, α_P3, and β_P1, respectively (Table 2). The data was shown in Figure 5 and Figure 6. Among the four tested peptides, the most promising peptide is α_P2, which has the nanomolar range inhibition. Confirmation of inhibition by α_P2 was achieved via Western blotting and transmission electron microscopy (TEM). Figure 6 shows that the degree of polymerization progressively decreases with increased concentrations of α_P2. The Western blotting intensities give the approximate IC_50_ value between 0.31 and 1.25 μM, which is consistent with the IC_50_ value from inhibition assays. Subsequently, TEM analysis confirmed the inhibitory effect by α_P2. Figure 7 shows the results of the TEM analysis, indicating that the inhibition of microtubule assembly arises from the inhibitory action of the peptide whereas the polymerization is not affected by the absence of α_P2. These observations strongly suggest that the identified candidate peptide, α_P2, inhibits *P. capsici* microtubule formation.

## 3. Discussion

Phytophthora blight caused by *P. capsici* is a devastating plant disease that affects many plants of economic importance in many parts of the world, especially in the countries of Europe, North America, and Asia. Several strategies have been adopted for controlling this disease, and yet extensive use of fungicides has led to the emergence of resistance in many oomycete pathogens including *P. capsici*. The extensive use of small molecule-based antifungal agents has had an adverse effect on human health and has been associated with loss of microbial diversity. 

As a result, there is an urgent need to develop effective, biologically safe and ecofriendly agents to control this disease. Thus, peptide-based inhibitors of *P. capsici* may represent an attractive strategy for plant protection. Peptides developed by the phage display method possess numerous advantages over nonpeptide-based antifungal agents. Since these peptides generally range in length from 7 to 15 amino acids, they can interact with their target protein with high specificity and affinity, an important requirement for the ideal therapeutic probe. Furthermore, these peptides are biologically safe as therapeutic and diagnostic probes.

In this study, we used the phage display method with M13 bacteriophage library to develop novel peptides that bind specifically to *P. capsici* α, β-tubulins. The library used in this study consists of DNA fragments fused to the gpIII minor coat protein of the M13 phage. As we screen for *P. capsici* α, β-tubulin-specific peptides in this library, the complexity can be 2.7 × 10^9^. Among the possible complexity, we expect to have some selective peptides with high affinity. After five rounds of biopanning, we successfully isolated 15 representative peptides possessing high affinity (picomolar range) for the α, β-tubulins. Among the 15 identified peptides, the performance of the four peptides α_P1, α_P2, α_P3, and β_P1 superseded the other peptides in terms of the binding affinity in the very low nanomolar range. Discrepancies in binding affinity values between the peptides of phage display and synthesized peptides are very common, but the four synthesized peptides possess reasonably high affinity. 

We also evaluated the cross-reactivity and selectivity of the identified candidate peptides toward *P. capsici* and homologous *H. sapiens* α, β-tubulin proteins. Interestingly, the identified candidate peptides did not show any cross-reactivity or specificity toward *H. sapiens* α, β-tubulin, indicating these peptides possess great potential as therapeutic agents against Phytophthora blight. Many of the fungicides that have lower selective toxicity toward mammalian tubulin also exhibit a lower binding affinity. One well-studied antifungal agent, benomyl, shows selective toxicity towards fungal tubulin rather than mammalian tubulin. This difference in toxicity is due to the different binding affinities seen within each species [24,25,26]. We demonstrated that the inhibitory potential of the candidate peptides against microtubule formation and the microtubule assembly using spectroscopic, Western blotting and TEM analyses. Interestingly, the IC_50_ value of the best candidate peptide, α_P2, was in the low micromolar range, which is within a similar range to that of the well-known inhibitor, benomyl. Like benomyl, we proposed that our screened peptides may also be prospective pesticide candidates.

## 4. Materials and Methods

### 4.1. Fungi, Bacterial Stains, Plasmid and Phage Library

*Phytophthora capsici* was provided by the Korean Agricultural Culture Collection (KACC). The fungi culture media—V8 vegetable juice agar—was purchased from Campbell Soup Co. *E. coli* BL21 (DE3; Novagen, Madison, WI, USA) and plasmid pET28a (Novagen) were used for gene expression. The M13 phage library (Ph.D.-12) screening kit (Ph.D. Phage Display Peptide Library kit) was obtained from New England Biolabs (Ipswich, MA, USA). All other chemicals used were of analytical grade and were purchased from Sigma (Sigma Aldrich, St. Louis, MO, USA). 

### 4.2. Expression and Purification of P. Capsici α, β-Tubulins

The expression and purification of α, β-tubulins of *P. capsici* and *H. sapiens* were purified using the method described by Koo et al. [26]. The purity of α, β-tubulins was analyzed by SDS-PAGE, resolved on a 4% stacking and 12% resolving mini gel (BioRad, Hercules, CA, USA). Proteins were visualized with Coomassie Blue R-250 stain, and their molecular weights were estimated with broad range precision standards (Sigma Aldrich, St. Louis, MO, USA).

### 4.3. Peptide Screening and Sequencing Analysis

For biopanning, the M13 Phage with 2.7 × 10^9^ peptide complexity was used. Briefly, 5 μg of α, β-tubulin was immobilized onto a polystyrene plate (SPL) via hydrophobic interactions, and the wells were then covered with 2% bovine serum albumin (BSA; 2% BSA, 50 mM Tris-HCl pH 7.5, 150 mM NaCl, and 0.1% Tween-20) for 1 h at room temperature. The wells were washed five times with wash buffer A (50 mM Tris-HCl pH 7.5, 150 mM NaCl, and 0.1% Tween-20) prior to use for biopanning. A specified amount of phage was added to the wells and incubated for 1 h at room temperature with gentle shaking. Wells were washed 10 times with buffer A and bound phages were eluted in chemical elution buffer B (0.2 M glycine-HCl (pH 2.2), 1 mg·ml^−1^ BSA). The eluents were neutralized with buffer C (1 M Tris-HCl, pH 9.1). In subsequent rounds of biopanning (second to fifth rounds), the concentration of detergent and salt in wash buffer A was increased as follows: 0.3% Tween-20 (second round), 0.5% Tween-20 (third round) and 0.5% Tween-20 plus 500 mM NaCl (fourth and fifth rounds). In the final round, bound phages were eluted by addition of a 10 M excess of α, β-tubulins. Eluted phages from each panning round were infected into *E. coli* ER2738 cells for amplification. After being propagated for 5 h, the bacterial cells were removed by centrifugation at 10,000 rpm for 10 min. The phages were partially purified by precipitation of the culture supernatant with polyethylene glycol (PEG, 20% (*w*/*v*) polyethylene glycol-8000 and 2.5 M NaCl) according to the manufacturer’s protocol. After centrifugation at 10,000 rpm for 40 min, phage pellets were resuspended in TBS buffer (50 mM Tris-HCl pH 7.5, and 150 mM NaCl) for use in the next round of biopanning. The DNA sequencing of single-strand phages was determined with an automatic sequencing service (Macrogen, Inc., Seoul, Korea) using the -96pIII sequencing primer. The amino acid sequence of the peptides was deduced from the genetic code information supplied by New England Biolabs.

### 4.4. Characterization of Phages

Phages displaying α, β-tubulin binding peptides were prepared, and their concentrations were determined by measurement of plaque-forming units (pfu) per volume. The binding affinities of phages displaying α, β-tubulin binding peptides were determined by the ELISA method, as described previously [22]. Briefly, α, β-tubulin binding peptides were added in a concentration-dependent manner to a 96-well plate containing α, β-tubulin protein (5 μg per well), as indicated. The wells were then washed 10 times with 1X TBST (50 mM Tris-HCl, pH 7.5, 150 mM NaCl, and 0.1% Tween-20), followed by probing with an anti-M13 antibody (mouse monoclonal antibody, 1:5000 in 1X TBST, 0.2 μg·ml^−1^; Amersham Bioscience, Little Chalfont, UK) for 1 h at room temperature. After washing five times with 1 X TBST buffer, anti-mouse IgG-horseradish peroxidase (HRP) conjugate was added, and the mixture was incubated for 1 h, followed by a color-developing reaction with tetramethylbenzidine (TMB) substrate solution ([TMB]/H_2_O_2_; Chemicon, Billerica, MA, USA). After 15 min, the reaction was terminated by the addition of 1 M H_2_SO_4_, and the optical densities were measured at 450 nm with a Biotrak multiwall plate reader (Amersham Bioscience). 

### 4.5. Evaluation of Binding Affinity and Specificity

Peptides were synthesized by AnyGen Inc. (Gwangju, Korea). To prevent unintended interaction with other molecules, the peptides used in this study were modified by acetylation at the N-terminus and amidation at the C-terminus. With the addition of a lysine residue, the peptides were labeled with FITC at the C-terminus. This labeling was used for the identification of the target binding property by means of fluorescence detection. The purity of the synthesized peptides was greater than 90%, as assessed by HPLC. The peptides were further characterized to estimate their binding affinity and specificity. Briefly, serial dilutions of FITC-labeled peptides were incubated with α, β-tubulin proteins immobilized on a 96-well plate for 2 h. The binding affinities of the FITC-labeled peptides were determined by a fluorescence microplate reader (Molecular Devices, Sunnyvale, CA, USA) using the soft Max V5 system, with excitation at 495 nm and emission at 520 nm [22,27]. The specificities of identified peptides for the target proteins were evaluated using the same fluorometric method described above, incorporating homologous *H. sapiens* α, β-tubulin. BSA protein was used as a negative control. 

### 4.6. Polymerization and Inhibition Assay

The polymerization and inhibition of microtubule assembly were studied using a spectroscopic method. Purified α, β-tubulins were mixed with 120 μL of polymerization buffer (80 mM PIPES (pH 7.4), 1 mM EGTA, 1 mM MgCl_2_, and 5% *v*/*v* glycerol) to a final concentration of 0.5 mg/mL total tubulin in 300 μL final reaction mixture. The polymerization assay was initiated with the addition of 2.5 mM GTP, and the reaction was continued until it reached the saturation point for 40 min. The contents were mixed, and the resulting turbidity due to microtubule formation was measured at 350 nm using an Optizen 2120 UV spectrophotometer. The inhibitory effect of the screened phages and peptides was tested under similar assay conditions, except with the addition of various concentrations of phage or peptide. Furthermore, the polymerization and inhibition of microtubule formation were also confirmed through sodium dodecyl sulfate–polyacrylamide gel electrophoresis (SDS-PAGE), Western blot, and transmission electron microscopy (TEM). For the TEM analysis, samples were prepared according to the protocol reported by Vulevic et al. [28]. Briefly, 30 μL of the saturated product of the polymerization reaction was immediately diluted with 10 μL of 0.4% glutaraldehyde, and the mixture was incubated for 3 min at room temperature. Then, 10 μL of the solution was applied to a 200 mesh, copper/formvar-coated grid (EMS) for 1 min, which was sequentially washed using two drops of dH_2_O and stained for 2 min with a drop of 2% uranyl acetate. Excess stain was removed by blotting with filter paper. Samples were air-dried and later viewed under a TEM.

## 5. Conclusions

Our study reports the development of novel peptides with high affinity for *P. capsici* α, β-tubulin which is a crucial biomarker in the control of *P. capsici* infection. The identified peptides showed great potential by inhibiting microtubule formation with IC_50_ values in the low micromolar range. In addition, the *P. capsici* α, β-tubulin-directed candidate peptides demonstrated higher binding affinity and selectivity toward the target proteins compared with the homologous *H. sapiens* α, β-tubulin. Moreover, our screened peptide, particularly α_P2, dissolved well in water due to the hydrophilic sequence and exhibited potent inhibition in tubulin depolymerization. These results indicate that, by targeting *P. capsici* α, β-tubulin, these peptides are promising candidates in the development of novel antifungal agents against Phytophthora blight. In future studies, it is recommended that the truncation of the peptide sequence be optimized and that live blight-infected plants be treated to determine the in vivo effectiveness of this peptide-based antifungal agent. It is expected that this peptide may protect crops from damage by blight in an ecofriendly manner, unlike the existing chemical agents currently used.

## Figures and Tables

**Figure 1 ijms-20-02641-f001:**
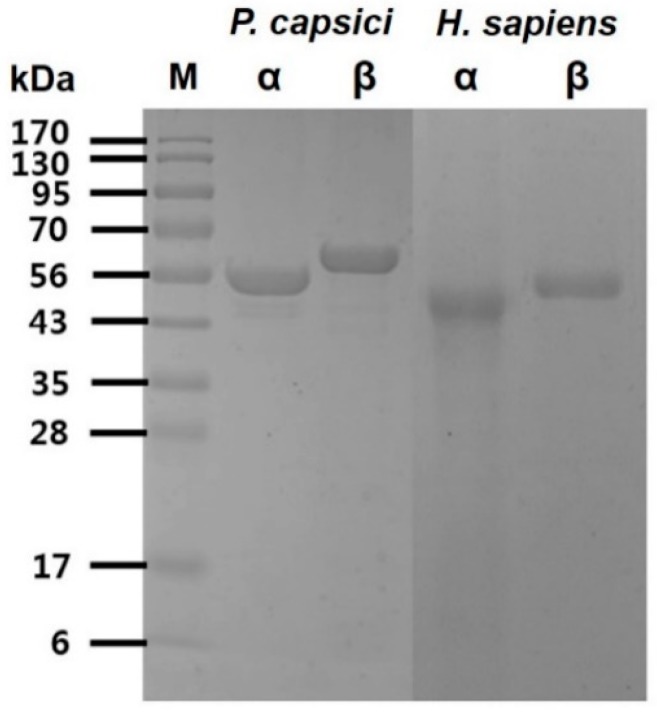
The SDS-PAGE analysis of the purification of *P. capsici* and *H. sapiens* α, β-tubulin. Electrophoresis was performed using a 12% polyacrylamide gel, and the staining of proteins was carried out using Coomassie blue R-250.

**Figure 2 ijms-20-02641-f002:**
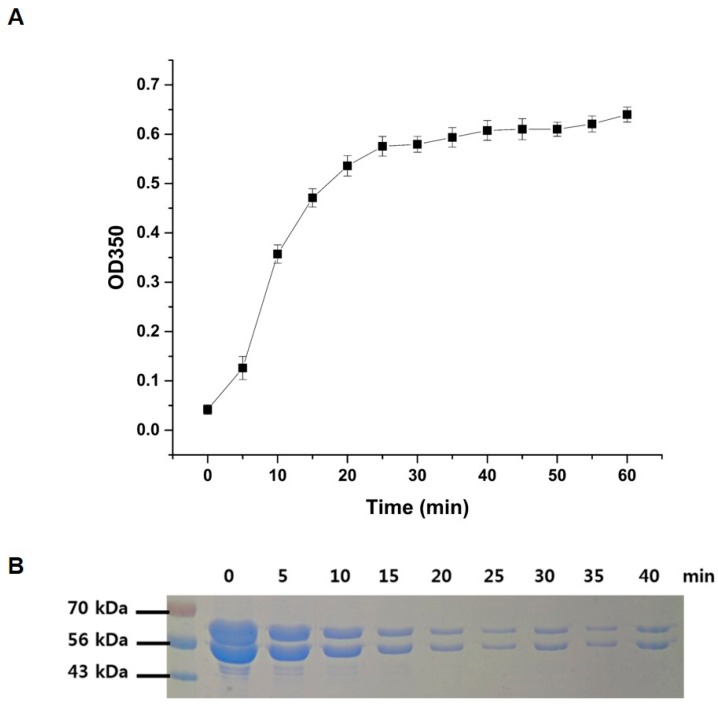
Polymerization of *P. capsici* α, β-tubulin. (**A**) The rate of the polymerization reaction was monitored by light scattering at 350 nm; as the polymerization progresses, the turbidity of reactant increases. (**B**) Time-dependent analysis of polymerization was confirmed by SDS-PAGE analysis. Due to the combining of α, β-tubulin for polymerization, both tubulins are consumed and the band intensity at 55 and 57 kDa decreases over time.

**Figure 3 ijms-20-02641-f003:**
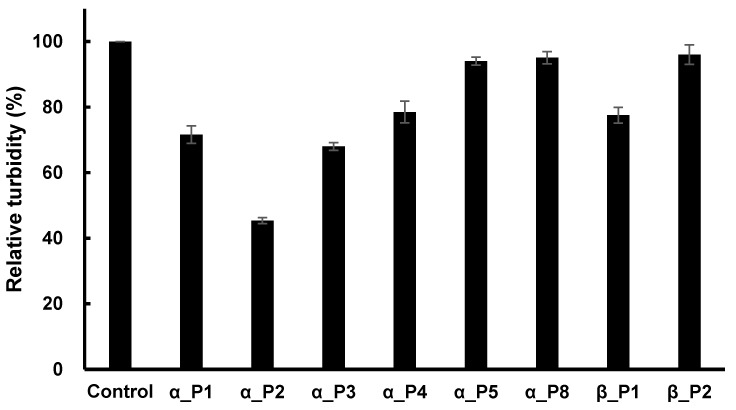
The relative activity of microtubule polymerization of *P. capsici* as a function of the fixed concentration of inhibitory phages. A fixed concentration of each phage (10 nM) was treated initially with the polymerization assay mixture for 40 min of reaction time. The percent inhibition of the peptides was as follows; α_P1, 28%; α_P2, 54%; α_P3, 32%; α_P4, 19%; α_P5, 5%; α_P8, 4%; β_P1, 21%; β_P2, 2%.

**Figure 4 ijms-20-02641-f004:**
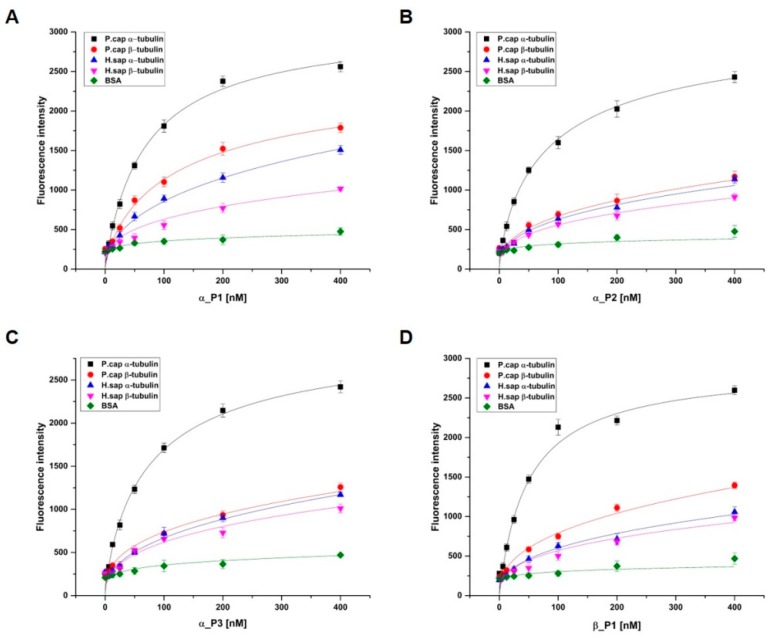
Determination of peptide specificities (**A**) α_P1, (**B**) α_P2, (**C**) α_P3, and (**D**) β_P1. Binding reactions were performed with a constant concentration of 2% BSA, 5 μg each of α, β-tubulin of *P. capsici* and *H. sapiens* in a 96-well white plate as a target and various concentrations of each peptide (0–400 nM). The specificities of the bound peptides were measured using a fluorometric method and were plotted against the concentration of the peptides used in the binding reaction. The data are representative of one of three independent experiments, and the specificities were obtained from nonlinear fitting of the saturation–binding curves.

**Figure 5 ijms-20-02641-f005:**
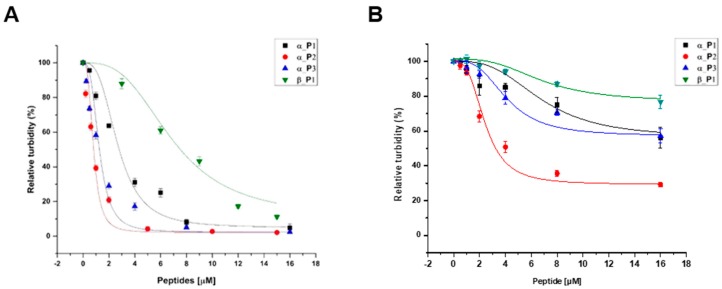
The relative activity of microtubule polymerization of (**A**) *P. capsici* and (**B**) *H. sapiens* as a function of increased concentration of inhibitory peptides. The activity for *P. capsici* yielded the following values of IC_50_; α_P1 (■), 2.69 μM; α_P2 (●), 802 nM; α_P3 (▲), 1.24 μM; β_P1 (▼), 6.91 μM.

**Figure 6 ijms-20-02641-f006:**
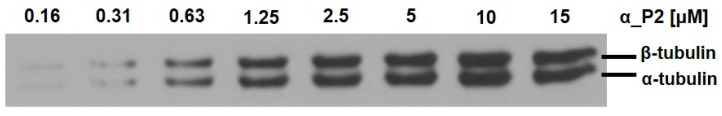
Western blot analysis to confirm the inhibition of microtubule polymerization using anti-His-antibody (rabbit monoclonal antibody, 1:1000 dilution in bovine serum albumin in Tris-buffered saline (TBST)). Reaction products were resolved on 12% SDS-PAGE, followed by Western blot analysis.

**Figure 7 ijms-20-02641-f007:**
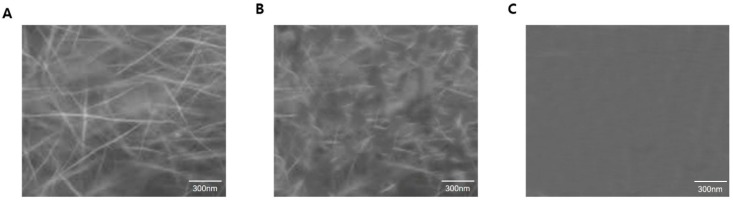
TEM images of polymerized *P. capsici* microtubules. α, β-tubulin were polymerized for 40 min. (**A**) Microtubules were observed without peptide. (**B**) Amorphous morphology of microtubules occurred predominantly with 1 μM of α_P2. (**C**) Ten micromolar α_P2 completely inhibited the formation of microtubules. As the concentration of peptide increased, the extent of polymerization was inhibited. The scale bar corresponds to 300 nm.

**Table 1 ijms-20-02641-t001:** *P. capsici* α, β-tubulin binding peptide sequences and phage binding affinities.

Type	Sequence	Frequency	K_d_ [pM]
α_P1	ITMSVPAHNAKE	39/169	34.2 ± 2.11
α_P2	TNTSWDPQYNPD	11/169	80.1 ± 1.13
α_P3	NHFVPTSNRFNA	11/169	120 ± 1.23
α_P4	NFTINGKTHRLW	6/169	298 ± 1.45
α_P5	NAITLLSPPLHK	5/169	57.1 ± 1.77
α_P6	SSHNHDSYHGTK	2/169	476 ± 2.73
α_P7	LMNPATMKTSSG	1/169	360 ± 1.87
α_P8	TNTSWDPQYNPD	1/169	47.6 ± 2.51
α_P9	SNMKPSMEYSSR	1/169	273 ± 1.79
α_P10	IGNSWPLTSHSW	1/169	144 ± 1.1
α_P11	SYNTFMYERASK	1/169	562 ± 1.69
α_P12	MVHSKASMWPGK	1/169	690 ± 2.27
α_P13	KVYAINSWTNYY	1/169	1200 ± 90.2
β_P1	TNPQARWHEYNF	61/112	41.9 ± 1.05
β_P2	NPIGDNYSGTGL	10/112	37.1 ± 1.41

**Table 2 ijms-20-02641-t002:** Binding affinity and inhibitory potency of the modified peptides against *P. capsici* and *H. sapiens* α, β-tubulin.

Type	Sequence	*P. Capsici*	*H. Sapiens*	*P. Capsici*	*H. Sapiens*
		K_d_ [nM]		IC_50_ [μM]	
α_P1	Ac-ITMSVPAHNAKEK(FITC)-NH_2_	69.5 ± 6.4	134 ± 48.8	2.69	N.D. ^a^
α_P2	Ac-TNTSWDPQYNPDK(FITC)-NH_2_	93.9 ± 11.2	256 ± 31.1	0.802	2.43
α_P3	Ac -NHFVPTSNRFNAK(FITC)-NH_2_	79.2 ± 6.8	162 ± 8.1	1.24	4.11
β_P1	Ac -TNPQARWHEYNFK(FITC)-NH_2_	49.7 ± 2.9	455 ± 50.7	6.91	N.D. ^a^

^a^ N.D. denotes not determined.

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
