# Peer review of "Novel Peptide-Based Inhibitors for Microtubule Polymerization in Phytophthora capsici"

_ijms, 2019, doi:10.3390/ijms20112641_

Round 1

Reviewer 1 Report

In order to find novel peptides with the ability to inhibit polymerization of tubulins found in P. capsici,  Lee et al. have screened peptides using M13 Phage library, and identified and characterized them for binding affinity, inhibition efficiency and specificity. They have found one peptide namely _P2 that exhibits specificity and very high inhibitory effect for P. capsica tubulins.

Considering the destruction that P. capsici can cause for various vegetable crops, the study is an important one and can be useful for developing better fungicides against P. capsici.  

The manuscript is well-written, and the results and interpretation are lucidly described. However, the authors’ claim that these peptides would be less harmful for human compared to the existing fungicides is not well-supported by the data. The manuscript should address the following comments.

Comments

Line 130, Table-2: It would be helpful if a similar table for Human tubulins is provided.

Line 135-137: Please describe how the fluorescence signal for each tubulin was distinguished in the mixture prepared to determine the specificities.

Section 2.5: Though binding affinity of the peptide towards tubulin was compared for human and P. capsici, their inhibitory effects were not. I would suggest including the latter. It is crucial for the peptides to have significantly higher IC50 for human tubulins, if these are to be considered as a potential fungicide.

Author Response

We would like to extend our gratitude to the reviewers for reviewing the manuscript. We also thank you for the positive responses from the reviewers. We explained our answers to reviewers’ comments thoroughly.  

Reviewer 1

In order to find novel peptides with the ability to inhibit polymerization of tubulins found in P. capsici, Lee et al. have screened peptides using M13 Phage library, and identified and characterized them for binding affinity, inhibition efficiency and specificity. They have found one peptide namely _P2 that exhibits specificity and very high inhibitory effect for P. capsica tubulins.

Considering the destruction that P. capsici can cause for various vegetable crops, the study is an important one and can be useful for developing better fungicides against P. capsici.  

The manuscript is well-written, and the results and interpretation are lucidly described. However, the authors’ claim that these peptides would be less harmful for human compared to the existing fungicides is not well-supported by the data. The manuscript should address the following comments.

Comments

1. Line 130, Table-2: It would be helpful if a similar table for Human tubulins is provided. 

- We revised the Table-2 by adding human tubulin data including binding affinity and inhibitory potency.

2. Line 135-137: Please describe how the fluorescence signal for each tubulin was distinguished in the mixture prepared to determine the specificities.

- To determine the probe specificity toward the target, we tested each protein with peptides and obtained fluorescence intensity. As compared each data, we analyzed binding property via cross-checking for each candidate probe to identify the specificity. 

3. Section 2.5: Though binding affinity of the peptide towards tubulin was compared for human and P. capsici, their inhibitory effects were not. I would suggest including the latter. It is crucial for the peptides to have significantly higher IC50 for human tubulins, if these are to be considered as a potential fungicide.

- We additionally tested peptide inhibitory potency toward human tubulin for directly comparison with P. capsicitubulin. The data are shown in table 2 and figure 5.

Reviewer 2 Report

This manuscript reported the discovery of a series of peptides which may potentially serve as anti-microtubule inhibitors. It started with the development of the antimicrotubule assay, followed by the screening of the peptides, and ended up with the inhibitory activity of the selected peptides assessed by various types of biochemical assays, including western blot, TEM, etc. Overall, it is a traditional chemical biology paper, with a focus of peptide binding and its inhibitory effect. The selling point of this manuscript resided in the nanomolar range activity of the final peptide. The experimental and result part is pretty straightforward to me, the things bothered me were the writing and clarity issue. Therefore, as far as I am concerned, this manuscript could be assigned as major revision.

First, in the introduction part, the background information is not detail enough. 1) Targeting microtubule is not a new topic, what was the previous effort from the literature? The author briefly mentioned the benomyl. How about others? 2) For selecting the peptides as novel therapeutics, only because it may provide good solubility and specificity? Why peptide? Was there any peptide used to serve as an anti-microtubule reagent?

Second, the results part, the author mentioned the screening of the peptides, however, it is not clear how those 15 sequences were generated. What is the M13 phage library? It is a commercial one or lab generated one. Without the library information, the readers will be confused, and do not understand where these sequences came from. In addition, how these peptides were synthesized, this information should be listed in the experimental part.

Finally, the conclusion part is missing, what is future direction and perspectives from this project, especially after obtaining such a valuable sequence.  

Author Response

We would like to extend our gratitude to the reviewers for reviewing the manuscript. We also thank you for the positive responses from the reviewers. We explained our answers to reviewers’ comments thoroughly.  

Reviewer 2

This manuscript reported the discovery of a series of peptides which may potentially serve as anti-microtubule inhibitors. It started with the development of the antimicrotubule assay, followed by the screening of the peptides, and ended up with the inhibitory activity of the selected peptides assessed by various types of biochemical assays, including western blot, TEM, etc. Overall, it is a traditional chemical biology paper, with a focus of peptide binding and its inhibitory effect. The selling point of this manuscript resided in the nanomolar range activity of the final peptide. The experimental and result part is pretty straightforward to me, the things bothered me were the writing and clarity issue. Therefore, as far as I am concerned, this manuscript could be assigned as major revision.

1. First, in the introduction part, the background information is not detail enough. 1) Targeting microtubule is not a new topic, what was the previous effort from the literature? The author briefly mentioned the benomyl. How about others? 2) For selecting the peptides as novel therapeutics, only because it may provide good solubility and specificity? Why peptide? Was there any peptide used to serve as an anti-microtubule reagent?

- As the reviewer mentioned, we revised the contents by supporting more information reading of previous researches for fungicides and why we use peptide as a novel anti-microtubule reagent including its strength in introduction and discussion parts.

2. Second, the results part, the author mentioned the screening of the peptides, however, it is not clear how those 15 sequences were generated. What is the M13 phage library? It is a commercial one or lab generated one. Without the library information, the readers will be confused, and do not understand where these sequences came from. In addition, how these peptides were synthesized, this information should be listed in the experimental part.

- The process for generating peptide sequence and peptide synthesis was described in section 2.3 and 4.5. However, we added more information up for easy to understand how to get it after biopanning. Furthermore, information of the M13 phage library was added in the Introduction and Materials and Methods part.

3. Finally, the conclusion part is missing, what is future direction and perspectives from this project, especially after obtaining such a valuable sequence.  

- We revised the conclusion by adding information for future directions and perspectives. 

Round 2

Reviewer 1 Report

The comments have been addressed and the manuscript can be accepted.

Reviewer 2 Report

The author addressed the questions and concerns I have, thus I agree to accept this manuscript to IJMS.